# Genetic Deletion of the *LINC00520* Homolog in Mouse Aggravates Angiotensin II-Induced Hypertension

**DOI:** 10.3390/ncrna9030031

**Published:** 2023-05-15

**Authors:** Xiaofang Tang, Chih-Hung Lai, Naseeb K. Malhi, Rahuljeet Chadha, Yingjun Luo, Xuejing Liu, Dongqiang Yuan, Alonso Tapia, Maryam Abdollahi, Guangyu Zhang, Riccardo Calandrelli, Yan-Ting Shiu, Zhao V. Wang, June-Wha Rhee, Sheng Zhong, Rama Natarajan, Zhen Bouman Chen

**Affiliations:** 1Department of Diabetes Complications and Metabolism, Arthur Riggs Diabetes Metabolism Research Institute and Beckman Research Institute of City of Hope, Duarte, CA 91010, USA; 2Cardiovascular Center, Taichung Veterans General Hospital, Taichung 40705, Taiwan; 3Institute of Clinical Medicine, National Yang Ming Chiao Tung University, Taipei 40705, Taiwan; 4Division of Chemistry and Chemical Engineering, California Institute of Technology, Pasadena, CA 91010, USA; 5Irell and Manella Graduate School of Biological Sciences, Beckman Research Institute of City of Hope, Duarte, CA 91010, USA; 6Department of Diabetes and Cancer Metabolism, Arthur Riggs Diabetes Metabolism Research Institute and Beckman Research Institute of City of Hope, Duarte, CA 91010, USA; 7Department of Bioengineering, University of California San Diego, La Jolla, CA 92093, USA; 8Division of Nephrology & Hypertension, University of Utah, Salt Lake City, UT 84132, USA; 9Veterans Affairs Salt Lake City Healthcare System, Salt Lake City, UT 84132, USA; 10Department of Medicine, City of Hope Comprehensive Cancer Center, Duarte, CA 91010, USA

**Keywords:** LEENE, hypertension, ECs, LncRNA

## Abstract

(1) Background: Hypertension is a complex, multifactorial disease that is caused by genetic and environmental factors. Apart from genetic predisposition, the mechanisms involved in this disease have yet to be fully understood. We previously reported that LEENE (lncRNA enhancing endothelial nitric oxide expression, transcribed from *LINC00520* in the human genome) regulates endothelial cell (EC) function by promoting the expression of endothelial nitric oxide synthase (eNOS) and vascular growth factor receptor 2 (VEGFR2). Mice with genetic deletion of the *LEENE/LINC00520* homologous region exhibited impaired angiogenesis and tissue regeneration in a diabetic hindlimb ischemia model. However, the role of LEENE in blood pressure regulation is unknown. (2) Methods: We subjected mice with genetic ablation of *leene* and wild-type littermates to Angiotensin II (AngII) and monitored their blood pressure and examined their hearts and kidneys. We used RNA-sequencing to identify potential *leene*-regulated molecular pathways in ECs that contributed to the observed phenotype. We further performed in vitro experiments with murine and human ECs and ex vivo experiments with murine aortic rings to validate the select mechanism. (3) Results: We identified an exacerbated hypertensive phenotype of *leene*-KO mice in the AngII model, evidenced by higher systolic and diastolic blood pressure. At the organ level, we observed aggravated hypertrophy and fibrosis in the heart and kidney. Moreover, the overexpression of human LEENE RNA, in part, restored the signaling pathways impaired by *leene* deletion in murine ECs. Additionally, Axitinib, a tyrosine kinase inhibitor that selectively inhibits VEGFR suppresses LEENE in human ECs. (4) Conclusions: Our study suggests LEENE as a potential regulator in blood pressure control, possibly through its function in ECs.

## 1. Introduction

High blood pressure (BP), or hypertension (HTN), is a worldwide epidemic and a major risk factor for cardiovascular diseases (CVD), including but not limited to heart failure, peripheral vascular disease, and stroke, as well as chronic kidney disease [1]. HTN is classified into essential or primary (accounting for 95% of all cases) and secondary types [2]. Essential HTN has no clearly identifiable causes but is largely thought to arise through a combination of factors, including genetic predispositions, lifestyle, and certain medical conditions (such as diabetes) [3]. Although pharmacotherapies have been shown to be effective in BP control in some patients with HTN, it remains challenging to achieve optimal BP control in others. In addition, many patients experience side effects from medications [4]. Therefore, there is an urgent need to better understand the molecular mechanism of HTN to develop targeted therapeutic treatment [5].

Accumulating data suggest a significant role of genetics in the development of HTN, as supported by the association between HTN and single nucleotide polymorphisms (SNPs) in genes related to BP regulation (e.g., angiotensinogen and angiotensin-converting enzyme [6,7]. However, genetics only explains a small portion of HTN [8,9], suggesting that other mechanisms, such as epigenetics, may be involved in the development of HTN. Identification of these epigenetic mechanisms underlying HTN can provide valuable information to develop improved treatment strategies for the management of HTN with higher precision.

The endothelial cell (EC) lining of blood vessels is key to BP maintenance. ECs control vascular reactivity and BP by releasing paracrine signaling molecules, such as nitric oxide (NO). Disruption of endothelial homeostasis alters the release of EC-derived factors (e.g., impaired NO production), leading to vasoconstriction and elevated BP [10]. Long noncoding RNA (lncRNAs), defined as non-coding RNAs comprising >200 nucleotides in length, have emerged to be important epigenetic regulators in EC biology and vascular function [11]. Numerous lncRNAs have been identified to mediate physiological and pathophysiological processes in the cardiovascular system [12,13]. However, research on the relationship between lncRNAs and HTN is still limited.

Several lncRNAs have been shown to be regulated by Angiotensin II (AngII) [14,15,16,17] or play a role in BP control [18]. For example, lncRNA Giver was found to be a partially conserved lncRNA in human, mouse, and rat vascular smooth muscle cells (VSMCs), and it is induced by AngII. Giver knockdown inhibited AngII-induced oxidative stress and proliferation in VSMCs. Another lncRNA, AK098656, also enriched in VSMCs, is only expressed in humans, and is upregulated in the plasma of hypertensive patients. AK098656-transgenic rats developed spontaneous hypertension, with increased VSMC synthetic phenotype and narrowed resistant arteries [19]. On the other hand, studies have also shown that AngII can regulate lncRNAs in ECs, which may, in turn, modulate EC viability, apoptosis, and migration [20,21]. However, it is not yet known whether the perturbation of an endothelial lncRNA can impact BP.

We have initially reported on the important role of a lncRNA that enhances endothelial nitric oxide synthase (eNOS) expression (LEENE) in the regulation of EC function [22]. Transcribed from the human *LINC00520* region, LEENE (NR025797.1) transcriptionally enhances the expression of eNOS and VEGF receptor 2 (vascular endothelial growth factor receptor 2, VEGFR2/KDR) and, in turn, promotes endothelial homeostasis and angiogenesis [23]. Mouse with genetic deletion of the *LEENE* homologous region (i.e., *leene*-KO) exhibited impaired hindlimb recovery after an ischemic injury in a diabetic model [23]. In the present study, we investigated the role of LEENE/leene in BP regulation and identified the potential underlying mechanisms.

## 2. Results

### 2.1. Leene-Knockout (KO) Mice Develop Higher BP in an Angiotensin II (AngII)-Induced HTN Model

To identify the role of LEENE in BP regulation, we used *leene*-KO mice (also referred to as KO mice hereafter), in which the syntenic region of human *LEENE/LINC00520* (mm39 chr14: 48,019,230–48,066,4410) was deleted using the previously described CRISPR-cas9 system [23]. Of note, this targeted region encodes three lncRNAs, i.e., Gm41148 (which is the putative mouse leene homolog), Gm49302 (a lncRNA embedded in the intron of *Gm41148*), and Gm35360 (a lncRNA divergently transcribed immediately upstream of *Gm41148*). According to the homology analysis, Gm41148 shares the highest conservation with human LEENE, and its expression levels are much higher than the other two lncRNAs in the heart and aorta [23]. Of note, there are two isoforms of leene (i.e., Gm41148-201 and 202), and we used a primer set that detects both isoforms. While the levels of Gm41148/leene were drastically decreased in the KO mice, the levels of Gm49302 and 35360 were not significantly different between the KO and their WT littermates (Appendix A). Under the basal condition, we did not observe a significant difference in BP or cardiac function in WT vs. KO mice [23] (Appendix A). However, when we challenged these mice to AngII infused subcutaneously through an osmotic minipump (Figure 1A), KO mice developed more severe HTN, evident by both higher systolic and diastolic BP, as compared to WT littermates (Figure 1B,C). The same experiments performed with mice of different ages (i.e., 2, 4, and 6 months old) generated consistent results, i.e., that *leene*-KO developed higher BP than WT mice (Appendix A). Furthermore, compared to no mortality observed in the AngII-infused WT animals (0 out of 24), AngII caused significantly higher mortality in KOs (8 out of 38) (Figure 1D).

### 2.2. Leene-KO Aggravates AngII-Induced Cardiac Hypertrophy and Pathology

As a major target organ of AngII, the heart undergoes hypertrophic and structural changes under AngII treatment, which may lead to cardiac failure [24,25]. In line with the aggravated HTN, KO mice showed a higher heart weight (Figure 1E–G), indicative of more severe cardiac hypertrophy. Of note, body weight was not significantly different between WT and KO mice (Appendix A). *Leene*-KO mice also exhibited worsened cardiac function, as revealed by a marked reduction in fractional shortening (%FS) and ejection fraction (%EF), along with no significant change in heart rate, as compared to the WT littermates (Figure 2A–D). At the histological level, compared to WT mice, the heart from *leene*-KO mice exhibited more pronounced hypertrophic and fibrotic phenotypes, characterized by the increased size of cardiomyocytes (Figure 2E,F) and Masson Trichrome staining (Figure 2G,H), respectively. In contrast, vascular density as marked by CD31 staining in the heart, was significantly decreased in KO mice (Figure 2I,J).

In line with these histological changes, several hallmark genes for pathological cardiac remodeling and fibrosis, including *BNP* (brain natriuretic peptide), *MYH7* (myosin heavy chain 7), and *COL1* (Type 1 collagen), were all increased in the heart from KO mice, as compared to the WT (Figure 2K). In contrast, hallmark genes governing EC homeostasis and maintaining physiological BP, such as NOS3 (which encodes) eNOS [26], *KLF2* (Krüppel-like factor 2) [27], *KLF4* (Krüppel-like factor 4) [28], and *ATP2B1* (ATPase Plasma Membrane Ca^2+^ Transporting 1), were decreased [29]. Moreover, genes that are known to promote HTN, such as NOS1 (which encodes inducible NOS/iNOS) [30] and *AGF* (Angiopoietin-Related Growth Factor), were increased in the KO hearts when compared with WT littermates (Figure 2K).

### 2.3. AngII Infused Leene-KO Mice Show Aggravated Kidney Damage

HTN is known to impact renal structure and function, which can exacerbate the hypertensive phenotype. Thus, we also performed histological studies on the kidneys of the AngII-infused mice. The glomerular size was larger in the kidneys of KOs than in the WT mice (Figure 3A,B). KO mice also exhibited more buildup of glomerular basement membrane (GBM), as indicated by PAS staining (Figure 3C,D) and increased collagen deposition in the kidneys (Figure 3E,F). Taken together, these results indicate aggravated renal damage and fibrosis in AngII-challenged mice due to *leene* deletion.

### 2.4. Potential Role of Endothelial LEENE in BP Regulation

We previously demonstrated that the level of leene is much higher in ECs as compared to non-ECs in various mouse tissues, including the artery and heart [23]. To further confirm this, we found that the leene levels were the highest in ECs as compared to cardiomyocytes, fibroblasts, and VSMCs in mice (Appendix A). Thus, to identify the potential mechanism underlying the observed hypertensive phenotype in *leene*-KO mice, we investigated the effect of *leene*-KO in the transcriptome of ECs using RNA-seq. To obtain enough RNA for this experiment, we isolated ECs from the lungs of the AngII-infused WT and KO mice. RNA-seq identified 235 differentially expressed genes (DEGs) due to *leene* deletion in ECs, with 102 up- and 133 down-regulated DEGs. While many immune pathways were enriched in the upregulated DEGs (e.g., *CXCR5*, *IL21R*, *CCR7* and *CD22*) (A,C), pathways such as “lipid and atherosclerosis’ and “fluid shear stress and atherosclerosis” were enriched in the downregulated DEGs (such as *NOS3*, *KLF2* and *KLF4*) (Figure 4B–D).

Next, we tested whether human LEENE (NR025797.1) could reverse the effect of *leene*-KO in ECs by overexpressing LEENE in lung ECs isolated AngII-challenged KO mice. LEENE overexpression (OE) robustly induced eNOS, KLF2 and KLF4 mRNA levels as compared to the Ad-GFP group (Figure 4E). Given the decreased vascular density in the heart of AngII-challenged KO mice, we also evaluated the effect of LEENE OE on the angiogenic capacity of ECs from these mice using an aorta ring assay. LEENE RNA significantly induced capillary outgrowth from the aorta isolated from the AngII-infused mice (Figure 4F), concomitant with the induction of eNOS, KLF2 and KLF4 (Figure 4G). Collectively, these data suggest a potential role of endothelial LEENE in the maintenance of normal BP.

### 2.5. Effect of TKIs on LEENE Levels in ECs

Hypertension is one of the most common cardiovascular comorbidity among cancer patients [31]. A frequent side effect of anti-cancer therapies, particularly with tyrosine-kinase inhibitors (TKIs), is hypertension. This is assumed to be due to TKIs targeting the vascular endothelial growth factor (VEGF) signaling pathway (VSP) [32]. Given the potential role of LEENE in BP regulation, we investigated the effect of TKIs on the levels of LEENE in human ECs (Figure 5A). Axitinib, a TKI used to treat advanced renal cell carcinoma with profound HTN as a side effect in over 50% of patients [31], strongly suppressed LEENE and eNOS expression, while inducing VCAM1 expression, both in HUVECs and in mouse aortic rings cultured ex vivo (Figure 5B,C). Moreover, axitinib also caused visible cell death. In contrast, lapatinib, another TKI used to treat advanced HER2-positive breast cancer without causing HTN as a side effect, showed an insignificant effect on EC viability and the levels of LEENE, eNOS, and VCAM1 RNA in vitro and ex vivo. These data suggest that the downregulation of LEENE may be one of the factors that contribute to the HTN induced by some TKIs, e.g., axitinib.

## 3. Discussion

In our previous studies, we found LEENE to be an essential lncRNA in the regulation of endothelial function, in part through enhancing the expression of eNOS and KDR [22]. In a diet-induced diabetic model, *leene*-KO mice exhibited impaired angiogenesis and tissue regeneration in response to hindlimb ischemia [23]. Herein, we provided evidence for the potential role of LEENE in the regulation of BP in the context of AngII-induced HTN. Specifically, knockout of *leene* in mice led to exacerbated HTN, accompanied by aggravated cardiac hypertrophy and fibrosis and kidney damage in an AngII-infused model. At the molecular level, ablation of *leene* downregulated genes essential for EC homeostasis and BP control and increased genes promoting cardiac dysfunction and HTN. Furthermore, the introduction of human LEENE RNA to murine ECs or aorta augmented the expression of multiple key genes contributing to normal EC function and physiological BP, e.g., eNOS, KLF2 and KLF4. Additionally, we provide a potential link between cancer therapy and HTN through LEENE. Collectively, these findings support a putative regulatory role of LEENE in BP control, in part through its action in ECs.

The effect of LEENE/*leene* in BP regulation seems to be context-dependent. As shown in our previous study, under basal conditions and a high fat andhigh sucrose diet, the BP of *leene*-KO mice was only modestly or insignificantly increased [23]. However, when challenged with AngII, the KO mice developed HTN. These findings suggest that LEENE/*leene* may be dispensable in the physiological state, but under a stress condition that changes vascular tone and causes cardiovascular and renal damage, LEENE/*leene* becomes necessary to maintain vascular function and BP. This is reminiscent of what we previously observed in the diabetic hindlimb ischemia model [23]. Namely, the *leene*-KO mice did not exhibit a different ischemic response under normal chow; however, when made diabetic, *leene*-KO showed a significantly impaired ischemic recovery and tissue regeneration, as compared to their WT littermates. These findings support the notion that lncRNA-mediated regulation can be context- and stress-dependent [33].

To provide a possible explanation for the observed hypertensive phenotype in the *leene*-KO mice, we performed transcriptome profiling of lung ECs isolated from these mice. Although lung ECs may not represent EC in other vascular beds and organs, the consistent decrease of eNOS, KLF2 and KLF4, i.e., key genes controlling EC and BP homeostasis in lung ECs and the heart due to *leene*-KO, complemented by the consistent increase of these genes by LEENE-OE in lung ECs and aortic rings suggest that LEENE/leene’s positive regulation of these genes is operative in ECs in various vascular beds. At the molecular level, we recently identified LEO1, a key protein component of RNA polymerase II-associated complex, to be a binding partner of LEENE. Silencing of LEO1 in ECs ablated the transcriptional induction of eNOS by LEENE. In ECs treated with AngII, the levels of LEENE and its interaction were decreased by AngII, suggesting that LEENE-LEO1 mechanism may be involved in the maintenance of physiological BP (Appendix A). Therefore, the ablation of this regulation likely contributes to the exacerbated EC dysfunction in the AngII model. Additionally, the decreased vascular density observed in the *leene*-KO hearts may also contribute to the exacerbated HTN, according to the existing literature [34].

Given that *leene*-KO is global in mice, we cannot exclude that LEENE/*leene* in other cell types may also participate in BP regulation. However, compared to ECs, the transcriptional activity of LEENE seems much lower in several other relevant cell types, e.g., cardiomyocytes, VSMCs, and kidney epithelial cells (Appendix A). It would be of interest to explore in future studies the regulatory role of LEENE in other cell types and its contribution to BP control.

Given the putative role of LEENE in BP regulation and HTN, we examined the effect of two commonly used cancer drugs on ECs, specifically the levels of LEENE. Interestingly, whereas lapatinib (a TKI that does not typically induce HTN) did not apparently alter LEENE, axitinib, a TKI that specifically targets VEGFRs and causes high incidence of HTN and EC dysfunction [35], substantially reduced LEENE levels in ECs. These data suggest that LEENE may underlie the cardiovascular toxicity of cancer drugs. As most, if not all, known drugs are designed to target protein-coding genes; there is a vast gap in knowledge on how these drugs may affect the non-coding genome. Acquiring knowledge on the link between lncRNA and HTN and how pharmacological interventions may impact lncRNAs will provide valuable insights into an improved understanding of the etiology of HTN as well as the potential effect of various therapeutics on BP.

In conclusion, the current study identified a potential role of lncRNA LEENE in the maintenance of BP and protection against HTN. There are limitations in this study. One of these is that only male mice were used to examine the effect of *leene*-KO in the regulation of BP, in part because female mice have been shown to be more resistant to AngII-induced HTN [36,37]. Another limitation is although there is partial conservation of LEENE between mice and men, as supported by similarities in the genomic context, nucleotide sequence, and expression patterns, it remains unclear to what extent the secondary structure and molecular function of mouse leene RNA resemble those of human LEENE. Future studies are warranted to address these limitations and unanswered questions.

## 4. Materials and Methods

### 4.1. Mouse Model

*Leene*-KO mice on the C57BL6 background were generated as previously described [23] and utilized in the present study. Male mice (ranging from 2 to 6 month-old) received implantation of Alzet osmotic minipumps (Lot#10419-21, Cat#Model 1004, Durect Corporation, Cupertino, CA, USA) subcutaneously in the dorsum of the neck under isoflurane anesthesia to maintain a delivery rate of 1000 ng/kg/min of AngII for 4 weeks or 2 weeks [17]. Two-week implantation was used for ex vivo experiments, whilst all other experiments were conducted at the 4-week mark. At the endpoints, the mice were euthanized with CO_2_ inhalation. 

### 4.2. Measurement of BP and Echocardiography

BP was measured using a noninvasive computerized tail-cuff system (Visitech, Apex, NC, USA) as previously described [38]. After the mice were placed in a plastic holder, the occlusion and sensor cuff were positioned at the base of the tail. All mice were given at least 1 week to adapt to the system prior to their BP measurement. BP was measured at least 20 times per mouse during the study period. Echocardiography was performed as previously described [39]. Briefly, mice under conscious condition were used with a Vevo 3100 Ultrasound Imaging System (FUJIFILM VisualSonics). Multiple parameters, including heart rate, fractional shortening (FS), and ejection fraction (EF), were determined from the ventricular M-mode tracing.

### 4.3. Histology and Immunostaining

Histological examinations were processed by the Pathology Core at the City of Hope. Heart and kidneys from mice were collected and fixed in 4% paraformaldehyde overnight. The fixed tissues were later dehydrated, sectioned into 4 µm paraffin slides, and subjected to Hematoxylin and Eosin (HE), Periodic acid-Schiff (PAS), or Masson’s trichrome staining. For immunohistochemistry (IHC), the slides were deparaffinized, rehydrated and incubated with endogenous peroxidase activity inhibitor and antigen retrieval reagents. IHC was performed on the Ventana Discovery Ultra IHC automated Stainer (Ventana Medical Systems, Roche Diagnostics, Indianapolis, IN, USA). Anti-mouse CD31 (Lot# 4, Cat#77699, Cell Signaling Technology, Danvers, MA, USA) was used at 1:200 dilution, followed by DISCOVERY anti-rabbit HQ (Cat#760-4815, Roche, Indianapolis, IN, USA) and DISCOVERY anti-HQ-HRP (Cat#760-4820, Roche, Indianapolis, IN, USA) incubation. The stains were visualized with DISCOVERY ChromoMap DAB Kit (Cat# 760-159, Ventana) and counterstained with hematoxylin. The IHC stained slides were digitalized and documented by NanoZoomer S360 Digital Slide Scanner (Hamamatsu, Bridgewater, NJ, USA) and viewed by NDP.view2 image viewer software (U12388-01, Bridgewater, NJ, USA).

### 4.4. Isolation of ECs from Murine Lungs and Aorta Ring Assay

Murine lung ECs were isolated as previously described [38]. For RNA seq, lungs were harvested from mice with Ang II implantation for 2 weeks and digested with Type I collagenase (Lot#41S21950, Cat#LS004196, Worthington Biochemical, Lakewood, NJ, USA). Sorting was done with Biotin-labeled anti-mouse CD144 (Lot#1321021038, Ca#130101983, Miltenyi Biotec, Auburn, CA, USA) and Anti-Biotin MicroBeads (Lot#5200210351, Cat#130090485, Miltenyi Biotec, Auburn, CA, USA) and MACS columns (Lot#5211018097, Cat#130042401, Miltenyi Biotec, Auburn, CA, USA). Sorted cells were collected and used for subsequent library preparation. For drug treatment, isolated murine lung ECs were cultured in six-well plates with EC growth medium (Cat#211-500, Cell Applications Inc. San Diego, CA, USA) Complete fresh media containing Ad-LEENE [23] was added to lung ECs at confluency. Ad-GFP was added as the vehicle control. Seventy-two hours post-treatment, the cells were harvested and used for subsequent experimentation.

The aortic ring assay was performed following a published protocol [40]. Briefly, thoracic aortas were dissected from isoflurane-euthanized mice after perfusion with cold PBS for 5 min. The fibroadipose tissue was removed, and the aortas were serially cross-sectioned into 1–2 mm rings. The aortic rings were singly put in 24-well plates and embedded in growth factor reduced Matrigel (Lot#1054001, Cat#356231, Corning, Chicago, IL, USA) for 10 min, then another 300 μL Matrigel was added and incubated for another 30 min. Then fresh EC growth medium was added to the plate. Following a 72-h incubation, complete fresh media containing Ad-LEENE or Ad-GFP was added into each well. Seventy-two hours post-treatment, the cells were harvested and used for subsequent experimentation. For drug treatment, aortas embedded in Matrigel were treated with axitinib (AG 013736, Cat#S1005, Selleck Chemicals, Houston, TA, USA) at 1 μM and lapatinib (GW-572016, Cat#S1028, Selleck Chemicals, Houston, TA, USA) at 5 μM for 24 h.

### 4.5. RNA Extraction, Quantitative PCR (qPCR), and RNA-Seq Library Preparation and Analysis

RNA was extracted from cells and tissues using TRIzol (Lot#391112, Cat#15596018, Invitrogen, Tewksbury, MA, USA) following the manufacturer’s instructions and was processed for qPCR and RNA-seq as previously described [38]. Briefly, the total RNA was reverse transcribed using the PrimeScriptTM RT Master Mix (Lot#Al82424A, Cat#RR036A-1, Takara, San Francisco, CA, USA), and cDNAs were used for PCR and qPCR analyses using Biorad CFX96. Each qPCR sample was performed in triplicate with iTaq Universal SYBR Green Supermix (Lot#64516552, Cat#1725121, BioRad, Hercules, CA, USA). β-actin (ACTB) and 36B4 were used as the internal controls for the in vitro and in vivo assays, respectively. For RNA-seq, 50 ng of total RNA per sample was used, and the libraries were prepared using the KAPA mRNA HyperPrep Kit (Lot#095613-1-1, Cat#08098123702, Roche Diagnostics, Indianapolis, IN, USA) following the manufacturer’s manual. The libraries were sequenced with HiSeq2500 using the SR50 mode. The annotation GTF file is GRCm38 (Ensembl release 84) for reference genome mm10. STAR [41] was used to align raw sequencing data to the hg38 genome, and Kallisto [42] was used to quantify transcript abundance in Transcripts per Million (TPM) values. DESeq2 [43] was then used to perform differentially expressed gene analysis with default parameters (adjusted *p*-values/false discovery rate <0.05 were considered significant). GO pathway enrichment analysis was performed through the Gene Ontology Consortium Platform, and Benjamini–Hochberg corrected *p*-values < 0.05 were considered significantly enriched pathways.

### 4.6. HUVEC Culture and Drug Treatment

HUVECs were purchased from and verified for negativity for mycoplasma contamination by Cell Applications, Inc. EC identity was authenticated by using immunostaining, flow cytometry, and the expression of CD144, CD31, and eNOS mRNA. HUVECs at passages 5–7 were cultured in M199 medium (Lot#1003369769, Cat#M2520-1L, Sigma, St Louis, MO, USA) supplemented with fetal bovine serum (Lot#2520294RP, Cat#10437-028, Gibco, Tewksbury, MA, USA), Sodium Pyruvate (Lot#2508976, Cat#11360-070, Gibco, Tewksbury, MA, USA), EC growth factor (Lot#1003484920, Cat#E1388-25UG, Sigma, St Louis, MO, USA), heparin sodium (Lot#SLCF6571, Cat#H3393-500KU, Sigma, St Louis, MO, USA) and Penicillin Streptomycin solution (Lot#30002285, Cat#30002CI, Coring, Chicago, IL, USA) under standard cell culture conditions (humidified atmosphere, 5% CO_2_, 37 °C). HUVECs were treated with axitinib (AG 013736, Selleck chem, Houston, TA, USA) at 1μM and lapatinib (GW-572016, Selleck chem, Houston, TA, USA) at 5 μM for 24 h. Twenty-four hours post-treatment, the cells were harvested and used for subsequent experimentation.

### 4.7. RNA IP (RIP)

Cells were UV-crosslinked (400 mJ/cm^2^), collected with cold PBS, and lysed with a buffer containing 50 mM Tris-HCl, pH 7.4, 100 mM NaCl, 1% NP-40, 0.1% SDS, 0.5% sodium deoxycholate ice for 15 min with protease inhibitor cocktail (Roche, Indianapolis, IN, USA). Protein A Dynabeads (Cat#10008D, Invitrogen, Tewksbury, MA, USA) incubated with IP antibody against LEO1 (Cat#A300-175A, Rabbit polyclonal, BETHYL, Milwaukee, WI, USA) or IgG (Cat#2729S, Rabbit, Cell Signaling Technology, Danvers, MA, USA) at room temperature for 2 h, then incubated with cell lysis at 4 °C overnight, followed by washes with a cold high-salt buffer. The RNA in the immunoprecipitates were extracted with Trizol.

### 4.8. Statistics

Statistical analyses for data other than RNA-seq (described above) were performed using GraphPad Prism. Unless otherwise indicated, two-group comparisons were performed using two-sided Student’s *t*-test and multiple-group comparisons were performed using ANOVA followed by Tukey’s test. The Kaplan–Meier estimate was used for the analysis of survival, with the log-rank test used to compare the difference. The *p*-values < 0.05 were considered statistically significant.

## Figures and Tables

**Figure 1 ncrna-09-00031-f001:**
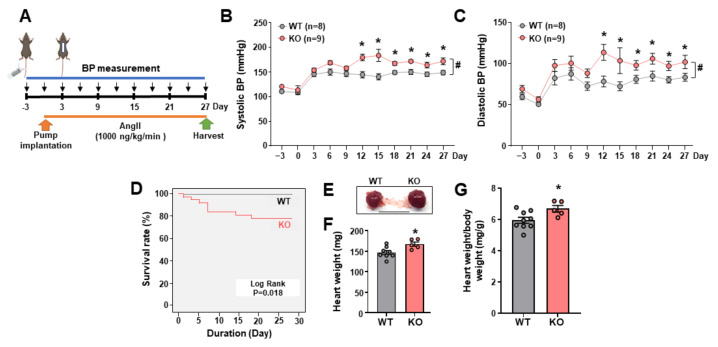
*Leene*-KO mice showed higher BP with AngII infusion. (**A**) Design of the Angiotensin-II (AngII)-induced hypertension (HTN) model: BP was measured every 3 days before and after osmotic pump implantation for 4 weeks. (**B**,**C**) Systolic and diastolic BP in 2-month-old mice with Ang II infusion (n = 8–9 mice per group). * *p* < 0.05 based on two-tailed Student’s *t*-test between WT and KO mice at the same time point. # *p* < 0.05 between WT and KO based on repeated-measures *t*-test. (**D**) Kaplan–Meier survival analysis in WT and KO mice receiving AngII (n = 24 vs. 38 per group). (**E**) Representative images of the heart obtained from AngII-treated WT and *leene*-KO littermates. (**F**,**G**) Heart weight and heart weight/body weight (mg/g) of WT and KO mice (n = 5–8 mice per group). Scale bar = 1 cm. Data are represented as mean ± SEM. * *p* < 0.05 between WT and KO based on Student’s *t*-test.

**Figure 2 ncrna-09-00031-f002:**
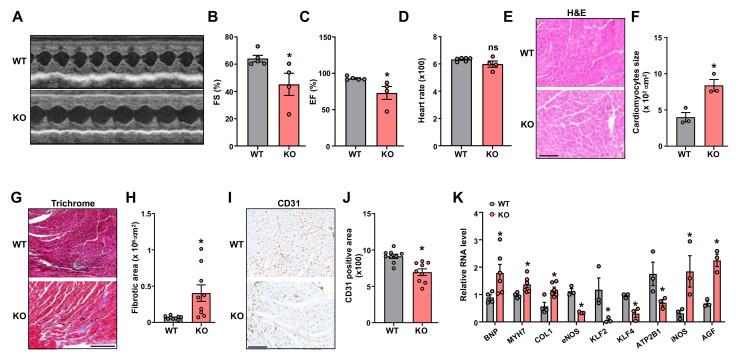
*Leene*-KO mice show impaired heart function in AngII model. (**A**) Representative M-mode images of echocardiography from WT and *leene*-KO mice with Ang II infusion on Day 27. (**B**–**D**) Fractional shortening (FS), ejection fraction (EF), and heart rate (n = 4–5 mice per group). (**E**,**F**) Representative images of H&E staining of heart. Quantification of the size of cardiomyocytes (n = 3 mice per group). (**G**–**J**) Representative images and quantification of Masson Trichrome staining (**G**,**H**) and immunohistochemical (IHC) staining of CD31 (**I**,**J**) in the heart. Three fields of view were randomly selected per animal, and three mice per group were quantified. (**K**) qPCR analysis of mRNA levels of genes as indicated in the heart, with 36B4 detected as the internal control (n = 3–6 mice per group). Scale bars = 100 μm (**E**,**I**) and 250 μm (**G**). Data are represented as mean ± SEM. * *p* < 0.05 based on two-tailed Student’s *t*-test.

**Figure 3 ncrna-09-00031-f003:**
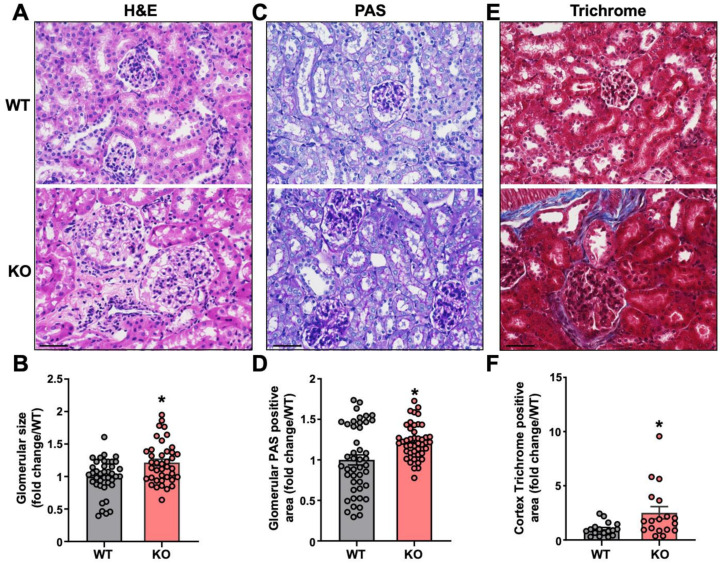
*Leene*-KO mice show aggravated kidney damage in AngII model. (**A**,**B**) Representative images of H&E staining of kidney and quantification of glomerular size. (**C**,**D**) Representative images of PAS staining and quantification of positive PAS staining area in the kidney. (**E**,**F**) Representative images of Masson Trichrome staining and quantification of the positive staining area in the kidney. Three fields of view were randomly selected per mouse and five mice per group were analyzed. Scale bar = 50 μm (**A**,**C**,**E**). Data are represented as mean ± SEM. * *p* < 0.05 based on two-tailed Student’s *t*-test.

**Figure 4 ncrna-09-00031-f004:**
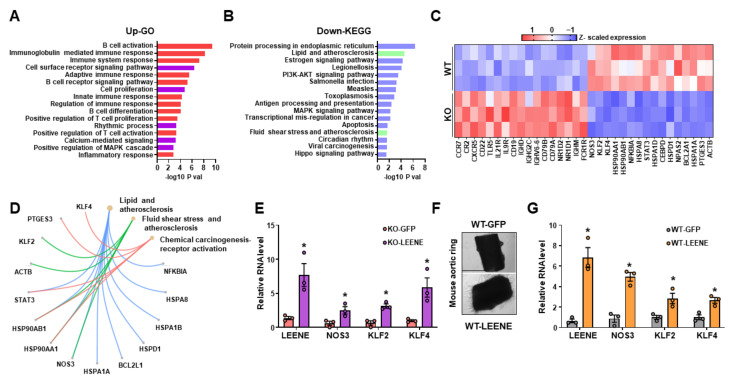
LEENE regulates pathways in ECs relevant to BP modulation. (**A**–**D**) WT and KO mice were infused with AngII for two weeks, and lung ECs were isolated from three mice per group for RNA-seq. (**A**,**B**) Top enriched biological pathways of upregulated (**A**) and downregulated (**B**) DEGs in lung ECs revealed by RNA-seq. (**C**) Heatmap showing the Z-scaled expression of select genes, including the upregulated DEGs involved in the immune and inflammatory pathways (indicated in red in A) and downregulated DEGs involved in flow response and atherosclerosis-related pathways (marked in green in B). (**D**) Downregulated DEGs involved in three select pathways. (**E**) qPCR analysis of indicated RNAs in lung ECs isolated from AngII-infused KO mice (for 2 weeks) and then treated with Ad-GFP/LEENE. (**F**,**G**) Aortic rings were isolated from AngII-infused WT mice and then treated with Ad-GFP/LEENE. Representative images are shown in (**F**), and qPCR analyses of indicated RNAs are shown in (**G**). n = three mice per group in (**E**,**G**). Scale bar = 1 mm. Data are represented as mean ± SEM. * *p* < 0.05 based on two-tailed Student’s *t*-test.

**Figure 5 ncrna-09-00031-f005:**
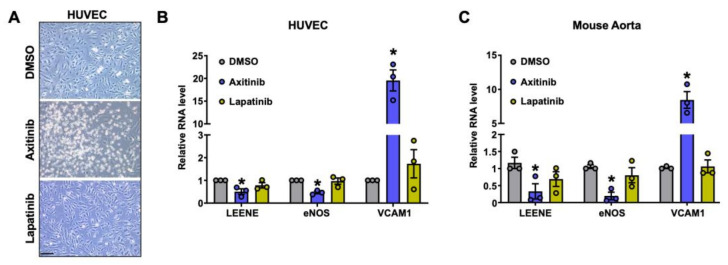
Effect of TKIs on LEENE levels in ECs. (**A**) Representative images of HUVECs treated with DMSO (vehicle), axitinib (1 µM) or lapatinib (5 µM) for 24 h. Scale bar = 100 μm. (**B**) qPCR analysis of indicated genes in HUVECs treated with DMSO, axitinib, or lapatinib. (**C)** qPCR analysis of indicated RNA level in WT aortic rings treated with DMSO, axitinib or lapatinib for 24 h (n = three mice per group). Data are represented as mean ± SEM from three independent experiments. * *p* < 0.05 between the indicated drug and DMSO group based on ANOVA followed by Tukey’s test.

## Data Availability

RNA-seq data from lung EC are available at GEO with accession No. GSE226638.

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
