# Peer review of "Genetic Deletion of the LINC00520 Homolog in Mouse Aggravates Angiotensin II-Induced Hypertension"

_ncrna, 2023, doi:10.3390/ncrna9030031_

Round 1
Reviewer 1 Report
Dear authors,
I have read with interest your paper. I find it a fairly conducted study that could be fit to be published.
Still, I cannot ignore the fact that you have changed the right order of the original article; I would advise you to put the chapters in this order: Introduction, Material and Methods, Results, and Discussions.
Also, I would like you to make a separate chapter ”Conclusions”.
Each study has its powerful points and some weak points. I think the authors should acknowledge that their study has some limitations.
Good luck!
Reviewer 2 Report
In this manuscript, the authors provide phenotypic analysis of the previously generated knockout mice of the target lncRNA. Although both in vitro and in vivo data are provided, the basic characterization of the target lncRNA is missing. Furthermore, the experimental evidence that overlapping and nearby lncRNAs are NOT affected are not provided, which raises a lot of questions about the phenotypes observed in this knockout mice that the authors previously generated. More specific comments are listed below:
Major points: [1] The authors should clearly write the basic information (e.g., chromosomal location, presence of isoforms) about murine homolog of LINC00520 in the main text. [2] After checking the authors previous publication [PMID: 36512424], the authors studied Gm41148. According to the latest annotation provided by the Ensembl database, this locus bears another lncRNA, Gm49302: https://www.ensembl.org/Mus_musculus/Gene/Summary?db=core;g=ENSMUSG00000115215;r=14:48028425-48042806 Thus, the authors must provide the experimental data regarding yet another lncRNA, Gm49302, so that the phenotypes observed in this study is NOT because of ablating Gm49302. [3] The target lncRNA is in the antisense direction to yet yet another lncRNA, Gm35360, according to the Ensembl database. The authors must provie the experimental data that the genetic deletion of the target lncRNA did not disturb the expression of Gm35360. If the expression of Gm35360 is disturbed in the knockout mice that the authors generated, further experimental data must be provided so that the the phenotypes observed in this study is NOT because of altered expression and function of Gm35360. [4] Lines 171 - 172: "We previously demonstrated that the level of leene is much higher in ECs as compared to non-ECs in various mouse tissues including the artery and heart (23).” In the previous study, the authors simply analyzed endotheial cells (EC) and non-EC. Thsu, no information is provided which cell types (e.g., cardiomyocytes, endothelial cells, smooth mucle cells, fibroblasts) the target lncRNA is expressed, if any. The authors could argue that the expression profiles of different cell types are provided in Supplemental Figure 4, but this is for human (not mouse) lncRNA. Thus, the authors must provide the results of RT-PCR from different cell types and in situ fluorescent hybridization (FISH) on the section of the murine heart. [5] No information about the mechanism of action of the target lncRNA is provided. The authors should perform RNA pull-down followed by mass spectrometry to identify protein binding partners and confirmation of binding by RNA immunoprecipitation. [6] The explanation for the usage of male mice only (instead of using both sexes) is missing. [7] Given that the total RNA was NOT treated with DNase I to digest the genomic DNA, the authors must provide the image of agarose gel with RT- sample after RT-PCR reaction for more than 40 PCR cycles. [8] Supplementary Table 1. eNOS mouse RT-PCR Forward: CTTGACCCAATAGCTGCTCAG Reverse: CACCTACGACACCCTCAGTG When the above primer sequences were used to run the UCSC In-Silico PCR program (https://genome.ucsc.edu/cgi-bin/hgPcr), the following results were obtained: The sequences and coordinates shown below are from GENCODE Genes, not from the genome assembly. The links lead to the Genome Browser at the position of the entire target sequence. >ENSMUST00000146326.7__Gm15587:2379+2856 478bp CTTGACCCAATAGCTGCTCAG CACCTACGACACCCTCAGTG CTTGACCCAATAGCTGCTCAGtgggtgaagggccctgggtgggccggctc tgtaacttccttggaaacaccagggatcccaagcagcgtcttgaggtaca gggcccatcctgaaccagaacaggggacacaaaggccaggatcaggtcat agtacaggatgagtgagctgtggttagttagcaagggcagagggaggcgg gctcgggaaagcaagggctcagagcagggctgataatggaaatgggatga gattggttgcttagggcttggggctcagggccaaaggcttgctgaagttc ttgcaatgccccttgatccctctggtccttcctggctttcactactctct gcagacaagggccaaatgtgaacaaagtggagtttcttggtttgggtcct gatctgggaccaggcctagaaacacgggcaggacctgtccttacctgctg agcctgggCACTGAGGGTGTCGTAGGTG Warning: this amplification is on the reverse-complement of ENSMUST00000030834.6__Nos3. >ENSMUST00000030834.6__Nos3:262-404 143bp CTTGACCCAATAGCTGCTCAG CACCTACGACACCCTCAGTG CTTGACCCAATAGCTGCTCAGtgggtgaagggccctgggtgggccggctc tgtaacttccttggaaacaccagggatcccaagcagcgtcttgaggtaca gggcccatcctgctgagcctgggCACTGAGGGTGTCGTAGGTG Warning: this amplification is on the reverse-complement of ENSMUST00000115090.5__Nos3. >ENSMUST00000115090.5__Nos3:262-404 143bp CTTGACCCAATAGCTGCTCAG CACCTACGACACCCTCAGTG CTTGACCCAATAGCTGCTCAGtgggtgaagggccctgggtgggccggctc tgtaacttccttggaaacaccagggatcccaagcagcgtcttgaggtaca gggcccatcctgctgagcctgggCACTGAGGGTGTCGTAGGTG As it is evident from the above, this primer is NOT specific to Nos3. Thus, the qRT-PCR result for Nos3 cannot be trusted. [9] ANP mouse RT-PCR Forward: CCTAAGCCCTTGTGGTGTGT Reverse: CAGAGTGGGAGAGGCAAGAC Similarly to the results of Nos3, the above primer pair is NOT specific to Nppa: The sequences and coordinates shown below are from GENCODE Genes, not from the genome assembly. The links lead to the Genome Browser at the position of the entire target sequence. >ENSMUST00000103230.4__Nppa:624+776 153bp CCTAAGCCCTTGTGGTGTGT CAGAGTGGGAGAGGCAAGAC CCTAAGCCCTTGTGGTGTGTcacgcagcttggtcacattgccactgtggc gtggtgaacaccctcctggagctgcggcttcctgccttcatctatcacga tcgatgttaaatgtagatgagtggtctagtgggGTCTTGCCTCTCCCACT CTG Warning: this amplification is on the reverse-complement of ENSMUST00000154192.1__Gm13054. >ENSMUST00000154192.1__Gm13054:169-321 153bp CCTAAGCCCTTGTGGTGTGT CAGAGTGGGAGAGGCAAGAC CCTAAGCCCTTGTGGTGTGTcacgcagcttggtcacattgccactgtggc gtggtgaacaccctcctggagctgcggcttcctgccttcatctatcacga tcgatgttaaatgtagatgagtggtctagtgggGTCTTGCCTCTCCCACT CTG [10] According to the latest annotation provided by the Ensembl database, there are 12 transcripts (isoforms) of LINC00520: https://www.ensembl.org/Homo_sapiens/Gene/Summary?db=core;g=ENSG00000258791;r=14:55781132-55796731 Which isoform(s) did the authors study? Why this isoform but not the others? This is especially important as the primer pair used does not detect all isoforms of LINC00520: LEENE human RT-PCR Forward: TTTCCCTCTTTGGGGTCTCA Reverse: GCCCTTTGATGAGTGAGTCG [11] Similar to [10], there are two isoforms of leene (Gm41148): https://www.ensembl.org/Mus_musculus/Gene/Summary?db=core;g=ENSMUSG00000115215;r=14:48028425-48042806 This information must be stated clearly as well as the information about each isoform, including whether the primer pair used detects both isoforms or not. Minor points:
 (1) The catalog and batch numbers of each chemical/reagent/kit must be listed. (2) The information about annotation file (i.e., GTF file with the version) is missing.
Reviewer 3 Report
In the submitted manuscript "Genetic deletion of LINC00520 homolog in mouse aggravates 2Angiotensin II-induced hypertension" the authors have presented the continuation of their previous work on lncRNA LEENE. The authors have examined the significance of LEENE in the pathophysiology of hypertension. The study is well-designed and the manuscript is clear and easy to follow. The results are coherent and well-discussed. The one thing that would strengthen the paper is the proof or at least analysis of the translational potential of these findings to humans. It is known that lncRNAs are not very well conserved between rodents and humans. Are there any proofs of the homology of LEENE between mice and men? If yes, please comment on that in the final part of the discussion. If not, please disclose in the limitations.
Round 2
Reviewer 2 Report
The authors sufficiently address all of my concerns/comments/questions that I have no further comment to make.